# Real-Time Monitoring-Based Stability Analysis of an Extra-Large LNG Tank Roof Under Construction

**DOI:** 10.3390/s25082498

**Published:** 2025-04-16

**Authors:** Yong Yang, Tuanhai Chen, Kezheng Zhang, Yu Song

**Affiliations:** 1CNOOC Gas and Power Group Co., Ltd., Beijing 100028, China; yangyong2@cnooc.com.cn (Y.Y.); zhangkzh5@cnooc.com.cn (K.Z.); 2College of Artificial Intelligence, China University of Petroleum (Beijing), Beijing 102249, China; songyu@cup.edu.cn

**Keywords:** ultra-large LNG tank, real-time monitoring, large span structure, staged construction, nonlinear buckling, rise–span ratio

## Abstract

**Highlights:**

**What are the main findings?**
The stability of the steel roof in the LNG tank is controlled by the construction load.The roof rise–span ratio has the greatest influence on stability.

**What are the implications of the main findings?**
It is necessary to monitor the roof concrete pouring construction process.Increasing the rise–span ratio is an effective method to enhance roof stability.

**Abstract:**

The predominant failure mode of ultra-large LNG tanks is buckling. Current stability analysis methods for the roofs of these tanks face challenges, such as inaccurate buckling load simulations and on-site monitoring data scarcity. This article presents a novel method for analyzing buckling, systematically investigating the key factors and mechanisms affecting roof stability based on real-time monitoring during construction. Firstly, a method including the “element birth and death” technique is proposed for roof stability calculation, and its accuracy is validated through real-time monitoring data. Secondly, the roof stability at each construction stage is analyzed. Finally, the impact of three key structural parameters is explored. The results indicate that the “element birth and death” technique can accurately simulate roof stability under construction conditions. The roof plate thickness, beam cross-sectional dimensions, and rise–span ratio all have a positive influence on stability. Among the factors considered, the section size of the longitudinal beam and the rise–span ratio have the most significant impact on roof stability, followed by the section size of the circular beam. In terms of material consumption, enhancing roof stability by increasing the rise–span ratio is an effective option.

## 1. Introduction

With the continuous advancements in LNG tank technology, their capacity has been steadily increasing. In 2017, three LNG tanks with a gross capacity of 270,000 m^3^ each were commissioned at the Samcheok LNG receiving terminal in South Korea. By 2024, a total of 11 LNG tanks, each with a net capacity of 270,000 m^3^, were put into operation at the Jiangsu Binhai and Zhuhai Jinwan receiving terminals in China, making them the largest LNG tanks in the world. An LNG tank with a capacity of 270,000 m^3^ has a diameter of 96.4 m and a height of 65.2 m. Its internal volume is capable of holding three Boeing 747 aircraft, and when fully loaded with LNG, the total weight exceeds 220,000 metric tons. Even larger LNG tanks and determining their ultimate capacity have emerged as pivotal areas of research focus in the field of cryogenic storage. As these tanks are typically large-span, thin-shell structures, the primary limiting factor for their capacity is their structural stability, which encompasses the stability of both the tank wall shell and the steel roof [1,2,3,4].

Many scholars have studied tank stability. In their studies, Pan and Liang, Zhao et al., and Uematsu et al. [5,6,7] developed a finite element model for open steel tanks and explored methods for analyzing tank stability under external pressure using FEA techniques. Additionally, extensive research has been conducted on the buckling behavior of storage tanks under diverse loading conditions, including seismic [8,9] and wind loads [10,11,12], fire-induced thermal stress [13,14], and differential settlement [15,16,17]. Most of the studies mentioned above primarily focus on tank wall buckling. However, for full-capacity LNG tanks, the outer tank is typically made of concrete, while the steel roof is a beam–plate composite structure that is more susceptible to buckling. Therefore, LNG tank stability analysis should place greater emphasis on roof stability. Yan et al. and Zhu et al. [18,19] investigated the stability of steel members strengthened by connecting plates; this structural style is similar to that of steel roofs, contributing to tank roof stability. Cheng et al. [20,21] conducted extensive research on this problem, exploring the use of an ESLRR shell as a replacement for the reticulated shell–steel plate composite structure in their buckling analysis. This provided a novel approach for assessing the stability of LNG tank roofs. However, their method simplifies the buckling load and does not account for the construction load during the roof concrete pouring process. Since the steel roof must support the weight of uncured concrete, which is more than ten times its own weight during construction, it is highly susceptible to buckling if the pouring process is not properly controlled. Thus, the construction load during concrete pouring is a critical factor in steel roof stability. In response, both Zhai et al. and Zhang [22,23,24] studied the impact of construction loads on roof stability, but their investigations did not consider staged construction. Instead, they applied the construction load all at once, which does not align with actual engineering practices. Currently, no scholars have investigated the influencing factors and mechanisms contributing to roof stability under construction conditions, which is a key factor in determining whether LNG tank expansions are feasible.

Therefore, in order to provide a theoretical basis for the design and optimization of LNG tank steel roof structures, reducing construction costs and promoting the application and popularization of ultra-large LNG storage tank technology, this paper focuses on a 270,000 m^3^ LNG tank. It investigates and determines the influencing factors and patterns affecting roof stability. Firstly, a stability analysis model of the LNG tank roof under construction conditions is established. Then, 144 sensors are installed on the steel roof to monitor dynamic stress and displacement data throughout the concrete pouring process. Real-time monitoring data are used to validate the accuracy of the analysis model. Finally, based on the established model, the buckling load of the steel roof at each stage of the pouring process is calculated. By investigating the influence of critical structural parameters on roof stability, this study provides a technical foundation for the design of ultra-large LNG tanks.

## 2. FEA Methods for Roofs Under Construction Conditions

### 2.1. Numerical Simulation Methods

When using FEA to calculate LNG tank roof stability, accurately establishing the analysis model is crucial, as this directly affects the results accuracy. The tank roof buckling analysis model must address two key issues: (1) The steel roof is a beam–plate composite structure. Once hardened, the concrete becomes an integral part of the system and therefore it is crucial to accurately account for the interaction between the three components (roof beams, plates, and concrete) [20,21]. (2) As the concrete on the steel roof is poured in stages, its deadweight and strength evolve over time during the pouring process. How can the effect of the roof concrete under construction conditions be accurately simulated?

To address the beam–plate synergy problem and model the interaction between the beams, plates, and hardened concrete, we employed the multi-point constraint (MPC) technique in ANSYS 16.0. As beams, plates, and hardened concrete exhibit no relative displacement post-construction, their MPCs are set as “Bonded” in both tangential and normal directions. The MPC methodology represents an advanced FE modeling technique that enables kinematic coupling between non-conformal meshes without nodal coincidence requirements. When implementing bonded contact interfaces, this approach ensures displacement continuity while allowing 12–18% mesh size variations between connected components. In ANSYS, the beam element BEAM188 and shell element SHELL181 were used to simulate the beams and plates. When establishing the beam element, a third node was introduced to ensure full adhesion between the beams’ upper flange and the plates. The hardened concrete elements were connected to the same nodes as the plate elements on the bottom, as shown in Figure 1a. This modeling approach not only accurately simulates the synergy between roof beams, plates, and hardened concrete, but also prevents the occurrence of a “pseudo-buckling” mode.

To simulate and model the staged concrete pouring process, “birth and death” behavior is introduced in ANSYS. The numerical model categorizes concrete elements into distinct component sets based on their respective pouring stages, with each set visually differentiated by color coding, as seen in Figure 1b. Before the concrete pouring begins, the “EKILL” command is used to “kill” all the concrete components by setting their stiffness matrices to near-zero values. Once a pouring stage is completed and the concrete reaches a certain strength, the “EALIVE” command is used to “activate” the corresponding component by restoring element stiffness. During the construction process, the concrete has no rigidity and is represented only as a distributed load, as shown in Figure 1b. External loading, such as seismic and wind loads, is not considered, because the pouring duration for tank roofs is relatively short, the probability of seismic events is extremely low, and construction is scheduled during periods with low wind pressure, where wind loads are negligible compared to pouring loads.

To validate mesh independence, a mesh convergence analysis was conducted. Three mesh sizes (0.2 m, 0.15 m, and 0.1 m, with a global model dimension of about 100 m) were adopted for the model, with self-weight loads from the roof concrete casting applied to the simulation model. The total displacement contours derived from these analyses are shown in Figure 2. The results demonstrate that when the mesh size is refined from 0.2 m to 0.15 m and 0.1 m, the displacement contours maintain consistent distribution patterns. Numerically, the displacements increase by 2.18% and 2.47%, respectively, with negligible increments that fall within acceptable computational tolerances. This validates that the 0.2 m mesh size satisfies the accuracy requirements for numerical simulation. To optimize computational efficiency, the present study therefore adopts 0.2 m mesh discretization throughout the modeling process.

Poured concrete’s stiffness contribution to the entire roof structure is reflected by defining the concrete elastic modulus, which increases with age and can be calculated according to Equation (1) [25]:(1)$$E_{c}(t)=\beta_{1}\beta_{2}E_{c28}\left(1-\exp(-\varphi t)\right)$$
where *t* is concrete age, d; *E*_c28_ is the elastic modulus cured for 28 days under standard conditions, MPa; *φ* = 0.09; and *β*_1_ and *β*_2_ are the correction coefficients of elastic modulus corresponding to the content of fly ash and mineral powder in concrete, which can be selected according to Appendix B of reference [25].

### 2.2. Validation Based on Real-Time Monitoring Results

Numerical simulation reliability may be compromised by an absence of failure criteria and progressive damage modeling in the constitutive representations of metallic, cementitious, and fiber-reinforced composite systems [26]. Therefore, in order to validate the simulation model’s accuracy, the pouring process parameters of a 270,000 m^3^ LNG tank roof were monitored under real conditions. Compared to the dynamic monitoring of bridges, monitoring the roof pouring process is considered to be static and short term, making it more straightforward to implement [27]. The span and height of the tank roof were 94.2 m and 13.3 m, respectively, with a rise–span ratio of 1/7.08. The reticulated beam structure consisted of 9 circular and 144 longitudinal beams, all made of H-shaped steel. Except the 24 main longitudinal beams with dimensions of HN400 × 200 × 8 × 13, the rest were HN350 × 175 × 7 × 11. The roof plate thickness was 6 mm, and the thickness of the roof concrete in the uniform section was 500 mm. The roof beams and plate were carbon steel, with material properties as shown in Table 1.

The real-time tank monitoring flowchart is shown Figure 3. Sensors were installed at the 36 intersection points of the 0°, 90°, 180°, and 270° longitudinal beams and the nine circular beams, as shown in Figure 4. In addition, four sensors were placed at each intersection to measure the longitudinal and hoop displacements and stresses, resulting in a total of 144 sensors.

The sensors include strain gauges and displacement meters. The former are surface-mounted intelligent vibrating wire-type, welded onto the steel beams. These strain gauges have a measurement range of ±2500 µε (microstrain) and an accuracy of 0.5% F.S. (Full Scale). The displacement meter is a laser rangefinder capable of simultaneously measuring inclination and displacement, with an angular measurement accuracy of ±0.01° and a displacement accuracy of ±1 mm.

Each sensor is connected to a transmission line, which runs along the longitudinal beams, as shown in Figure 5. The transmission lines from all sensors are eventually gathered at a HUB located above the ceiling plate, as shown in Figure 6. The HUB is equipped with a wireless signal transmission device, allowing the stress and displacement data of the steel roof to be easily collected during the concrete pouring process.

The tank roof concrete is poured in five rings; the detailed pouring scheme is shown in Table 2.

The roof concrete pouring process is shown in Figure 7. After each roof ring is poured, a period of waiting time is required until the strength reaches the design requirements.

The entire process, including the pouring of five rings, lasted for 635 h, during which time the stress and displacement were monitored in real time. The results of beam stress during construction are shown in Figure 8.

After each roof ring is poured, the vertical displacement at each position is recorded. The displacement–time curve is shown in Figure 9.

To verify the accuracy of FEA methods, the stress and displacement data from the pouring of the first and second rings were selected for comparison. The comparison results are shown in Figure 10 and Figure 11, where the X-axis represents the horizontal projection distance from the tank center.

From Figure 10, it can be observed that the trend in the FEA curve closely matches the real-time monitoring curve, with a slightly smoother simulation result. The FEA values are slightly lower than the measured results, which may be caused by the inherent construction imperfections in the actual process. These defects result in reduced structural stiffness, amplifying stress–strain responses and displacement magnitudes. A comparative analysis reveals that the discrepancies between numerical simulations and field measurements can be quantified at 14.29% for stress distributions and 15.87% for displacement profiles, aligning with typical tolerances for large-scale structural monitoring projects.

As shown in Figure 11a, the concrete in the first ring has gained some strength by the time the second ring is poured, causing stress in pouring zone of the former (the outermost zone) and exhibiting rebound. In Figure 11b, the displacement in this zone decreases more slowly. Quantitative comparisons between numerical simulations and field monitoring results reveal discrepancy rates of 17.04% for stress and 18.81% for displacement, aligning with acceptable engineering thresholds for complex structural systems.

The comparison results above show that the change trends of the FEA and real-time monitoring results are consistent, and the values fall within the acceptable error range. This confirms the accuracy of both the FEA model and the simulation method.

## 3. Buckling Analysis Under Construction Conditions

### 3.1. Buckling Analysis Method

The tank roof buckling analysis methods include both linear and nonlinear buckling analyses. In the latter, the initial geometric imperfection of the structure must be considered. The location and size of this imperfection can be determined based on measured results or the lowest mode from a linear buckling analysis of the structure [28,29,30]. The maximum imperfection size is typically set to 1/300 of the roof span [31]. In ANSYS, the UPGEOM command can be used to introduce the initial imperfection to the model, as shown in Figure 12.

To ensure accuracy, both material and geometric nonlinearities must be considered in nonlinear buckling analysis. Material nonlinearity is accounted for by defining the constitutive relationships for steel and concrete, considering plastic yielding. Geometric nonlinearity is introduced by setting NLGEOM, ON.

Prior to computation, an initial load exceeding the critical buckling load must be estimated and applied to the model. In nonlinear buckling analysis, the general method involves gradually increasing the applied loads until the solution starts to diverge. It is important to use a sufficiently small load increment—approximately 1/200th of the estimated buckling load—as the applied load approaches the critical buckling load. If the load increment is too large, the predicted buckling load may not be accurate. The arc-length option should be activated to promote convergence. Finally, the load–displacement curve can be obtained, as shown in Figure 13, in which the load corresponding to the maximum displacement represents the buckling load.

### 3.2. Non-Stage Pouring Buckling Analysis

Tank roof stability is analyzed under the assumption that all the roof concrete is poured at once rather than in stages. In this case, the concrete does not contribute to the overall stiffness of the structure, and the entire weight of the concrete is applied as pressure on the roof plate. Due to the varying concrete thickness across the roof, the pressure applied to the plate differs accordingly. The pressure generated by the concrete’s deadweight is 43.2 kPa at the outside regions where it is thickest. At regions of a constant, 500 mm thickness, the pressure is 12.1 kPa.

In the nonlinear buckling analysis, five times the concrete self-weight load is applied for calculation, as shown in Figure 14a. Based on the nonlinear solution for this load, the vertical displacement contours of the tank steel roof after buckling are presented in Figure 14b.

As shown in Figure 14b, under buckling conditions, the vertical displacement at the outer edge (the regions of the eighth and ninth circular beams) is larger than at other locations. This result arises due to the higher construction loads (more than 30.2 kPa) in this region, which contribute to amplified displacements. However, the proximity to perimeter-fixed supports (within 5 m radial distance) effectively constrains buckling potential. However, upward displacement occurs at the center of the roof. The displacement direction is opposite to the applied load, indicating that buckling may occur in this region. As such, it is necessary to focus on this area when conducting buckling analysis. The load–displacement curves of the circular beams from the first to the ninth ring at 0° are shown in Figure 15.

As shown in Figure 15, when the load factor reaches 0.96, the load–displacement curves of the first and second circular beams undergo a turning point, indicating buckling. When the load factor reaches 1.2, the load–displacement curve of the third circular beam undergoes a turning point, indicating that the buckling load of this beam is 1.2 times the construction load. There is no turning point in the load–displacement curves of the fourth to the ninth circular beams, indicating that these six beams have not buckled. This analytical conclusion corroborates the findings above, as Circular Beams 1–3, located in the central roof region, exhibit higher buckling susceptibility due to their extended distance from perimeter fixed supports (radial span > 30 m); Circular Beams 4–9 (within 15 m radial distance), however, demonstrate inverse behavioral characteristics.

From the above analysis, the stability safety factor is 0.96, which does not meet the requirement that the safety factor should be greater than two [31]. This indicates that pouring all the roof concrete at once is not feasible for ultra-large LNG tanks. Although the staged pouring method takes more time, it is both reasonable and necessary, as demonstrated by its use in the actual project.

### 3.3. Staged Pouring Buckling Analysis

As demonstrated in the above analysis, it is essential to study the stability of staged pouring. The principle of phased pouring is to keep the volume of concrete poured each time as uniform as possible, with the pouring boundaries ideally located at the circular beam positions. The phased pouring scheme for a 270,000 m^3^ LNG tank is based on preliminary studies and shown in Table 2. In this scheme, the entirety of the roof concrete is poured in five rings. Due to the larger concrete thickness in the outer rings, the corresponding radial width is smaller, resulting in a higher concrete deadweight pressure.

The pouring process of each ring is simulated and the buckling load is calculated based on the staged pouring scheme. The load–displacement curve the concrete pouring of each ring is shown in Figure 16.

As shown in Figure 16, the load–displacement curve remains relatively smooth during the pouring of the first ring concrete. Buckling occurs when the load factor reaches 9.75. After buckling, the structure’s bearing capacity does not decrease sharply. As the load increases, the displacement decreases slowly, indicating that secondary buckling may occur. The safety factor for the buckling load is relatively large, and the structure remains relatively safe. As the pouring load expands toward the center of the tank, the load–displacement curve changes dramatically when the second, third, and fourth rings are poured. The structural system exhibits abrupt strength degradation, accompanied by accelerated displacement amplification, upon surpassing the critical buckling load threshold. Under these conditions, local buckling may occur, preventing further increase in load. The safety factors for the pouring of the three rings are 6.30, 5.41, and 4.41, respectively. When the fifth ring is poured, the concrete in the other poured regions has gained some strength. Buckling deformation mainly occurs in the central casting regions, where there is both holding pressure below the roof plate and concrete deadweight pressure above. When these two pressures are not balanced, the steel roof is prone to distortion and buckling. This buckling deformation has strong integrity, causing the load–displacement curve to remain relatively flat. However, the buckling load safety factor is 2.11, which is relatively low.

The buckling displacement contours of the pouring of each ring are shown in Figure 17.

As shown in Figure 17, when the first ring of concrete is poured, the maximum displacement occurs at the outside region, but this is small. When the third and fourth rings of concrete are poured, the vertical displacement increases significantly, and local buckling occurs in many zones. When the fifth ring of concrete is poured, noticeable steel roof warping occurs. The characteristics observed in the displacement contours are consistent with the load–displacement curves. Consequently, the central roof region exhibits reduced buckling resistance capacity, necessitating the rigorous verification of localized stability parameters during both design optimization and construction sequencing phases.

## 4. Parameter Sensitivity Analysis of Tank Roof Stability

The above investigation shows that the buckling load of the last ring of concrete pouring is the lowest. Although it meets the minimum code requirements, the margin is very small. If the tank capacity is further increased, it may not meet the requirements. Therefore, it is essential to explore methods to improve LNG tank roof stability. Chen et al. [1] studied the influence of construction parameters in this regard. This study will focus on analyzing the impact of structural parameters on roof stability, including the roof plate thickness, beam section size, and rise–span ratio.

### 4.1. Influence of Roof Plate Thickness

To investigate the influence of roof plate thickness on stability, the buckling load factor for thicknesses of 6 mm to 12 mm was calculated; a variation curve is shown in Figure 18.

As shown in Figure 18, with increased plate thickness, the buckling load factor gradually increases, but the rate of increase slows down. Furthermore, when the thickness is doubled, the safety factor only increases by 1.90%, which is much less than the increase in thickness. Therefore, it can be concluded that the roof plate thickness has some influence on the stability of the steel roof structure, but the effect is slight. This is because, in a steel roof beam–plate composite structure, stability is primarily controlled by the beam components, while the plate component mainly serves functional roles, such as acting as a formwork during concrete pouring and providing gas tightness during operation. Therefore, the roof plate design principally satisfies static load-bearing requirements for poured concrete deadweight, with a thickness of 6 mm proving structurally adequate. This thickness concurrently represents the economically optimized minimum gauge in fabrication processes.

### 4.2. Influence of Beam Section Size

When studying the influence of beam section size on structural stability, the different contributions of the circular and longitudinal beams were considered separately. In the analysis, the section dimensions of all the circular or longitudinal beams are multiplied by the corresponding amplification factor, which ranges from 1.0 to 1.5 in increments of 0.05. Curves showing the variation in the buckling load factor with the beam section size amplification factor are presented in Figure 19.

As shown in Figure 19, the tank roof buckling load factor increases with increased beam section size. By comparing the two curves, it is evident that the influence of the longitudinal beam size on structural stability is much greater than that of the circular beam. The stability of the structure improves continuously as the longitudinal beam section size increases; however, material consumption also increases with the amplification of the beam section size. As can be seen in Table 3, the buckling load factor and material consumption for different amplification factors of the longitudinal beam section increase.

As can be seen in Table 3, when the section size of the longitudinal beam is amplified by 50%, the buckling load factor increases by 129.88%, which is remarkable. However, at the same time, the steel consumption of the whole roof is increased by 47.3%.

However, the incremental enhancement of buckling load factor associated with circular beam cross-sectional magnification proves significantly less pronounced compared to that induced by longitudinal beam dimension amplification. When the magnification factor reaches 1.35, the structural stability remains unchanged, even with further increases in section size. This indicates that there is a limit to the influence of the circular beam size on overall structural stability.

The difference in the influence on structural stability between the circular and the longitudinal beam arises from the distinct roles they play in the steel roof structure system. The ends of the longitudinal beams are fixed on top of the concrete outer tank, supporting the entire roof structure, while the circular beams are placed between the longitudinal beams to prevent lateral deformation. Therefore, increasing the sectional dimensions of the longitudinal beams can rapidly enhance structural stability. Amplifying the sectional dimension of the circular beams, within a certain range, can improve their ability to resist the lateral deformation of the longitudinal beams, thus enhancing overall stability. However, once the sectional dimension of the circular beams exceeds a certain threshold, their capacity to prevent the lateral deformation of the longitudinal beams plateaus, resulting in no further improvement in structural stability.

### 4.3. Influence of Rise–Span Ratio

The tank roof rise–span ratio constitutes a critical parameter governing both structural stability and constructability. While this ratio significantly influences buckling resistance, excessive values (typically >1:4.5 per ACI 334.1R-92(2002) recommendations) introduce practical construction challenges. The rise–span ratio of the LNG tank roof discussed above is 1/7.08, and to analyze the impact of this ratio on structural stability, its maximum magnification factor is set as 1.6. The parametric response curve of buckling load factors, corresponding to rise–span ratio amplification coefficients ranging from 1.0 to 1.6, is presented in Figure 20.

Figure 20 illustrates that as the rise–span ratio increases, the buckling load factor also progressively increases. This is because, with increasing rise–span ratio, the angle of the roof arc relative to the horizontal plane becomes steeper, which reduces the vertical component of the pouring load on the roof. Since the vertical component of the pouring load is the primary factor causing roof buckling, the structural stability improves as the rise–span ratio increases. The increase in the buckling load factor and material consumption for different rise–span ratio amplification factors is presented in Table 4.

As shown in Table 4, the buckling load factor increases by 107.2% when the rise–span ratio amplification factor increases from 1.0 to 1.6, while material consumption only rises by 6.33%. The enhanced buckling resistance achieved through rise–span ratio optimization originates not from augmented structural stiffness, but rather from improved load redistribution via morphological configuration optimization. This geometric refinement enables more than 100% buckling capacity enhancement with merely 6–7% material increment. Therefore, increasing the rise–span ratio is a more efficient way to enhance structural stability compared to amplifying the section size of the longitudinal beams. However, if the rise–span ratio becomes too high, this may complicate the concrete pouring process. Thus, improving tank roof stability by increasing the rise–span ratio within a certain range is a more practical solution.

## 5. Conclusions

This study investigates analysis methods and parameter sensitivity in super-large LNG tank roof stability during the construction process; our analysis model was validated using real-time monitoring data from the world’s largest 270,000 m^3^ LNG tank. The main conclusions drawn are as follows:(1)For ultra-large LNG tanks, staged roof pouring is essential in order to meet stability requirements. Real-time monitoring data demonstrate that the FEA method, including the “birth and death” technique, can accurately simulate roof stability under staged pouring conditions.(2)The roof plate thickness, beam section size, and rise–span ratio all positively affect tank roof stability. Among these, the longitudinal beam section size and the rise–span ratio have the greatest impact on stability, followed by the circle beam section size, which has a limiting value. The roof plate thickness has a minimal effect on stability and can be considered negligible.(3)Taking into account both material usage and construction factors, appropriately increasing the rise–span ratio within a reasonable range is the optimal method for enhancing LNG tank stability.

The stability analysis method and influence patterns proposed in this study can be extended to even larger LNG tanks, which will significantly advance the technological development of LNG containment systems.

## Figures and Tables

**Figure 1 sensors-25-02498-f001:**
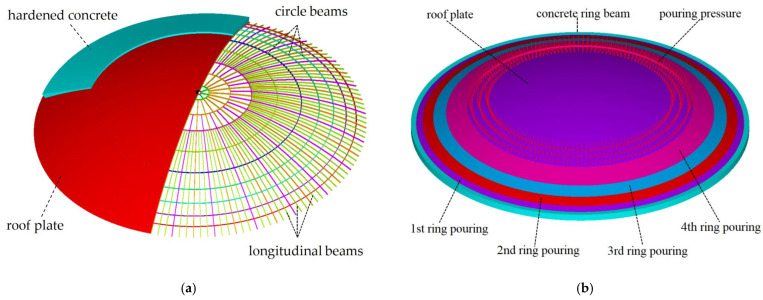
Finite element model for simulating the LNG tank roof construction process. (**a**) Roof beams and (**b**) roof plate and pouring rings.

**Figure 2 sensors-25-02498-f002:**
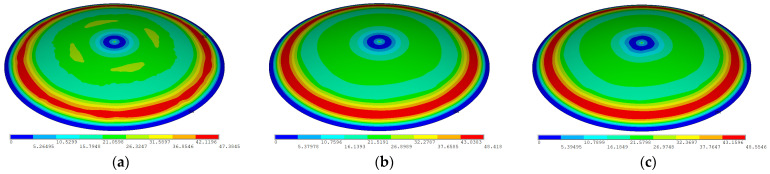
The displacement contours of steel roofs under concrete weight load. (**a**) Element sizes of 0.2 m, (**b**) 0.15 m, (**c**) and 0.1 m.

**Figure 3 sensors-25-02498-f003:**
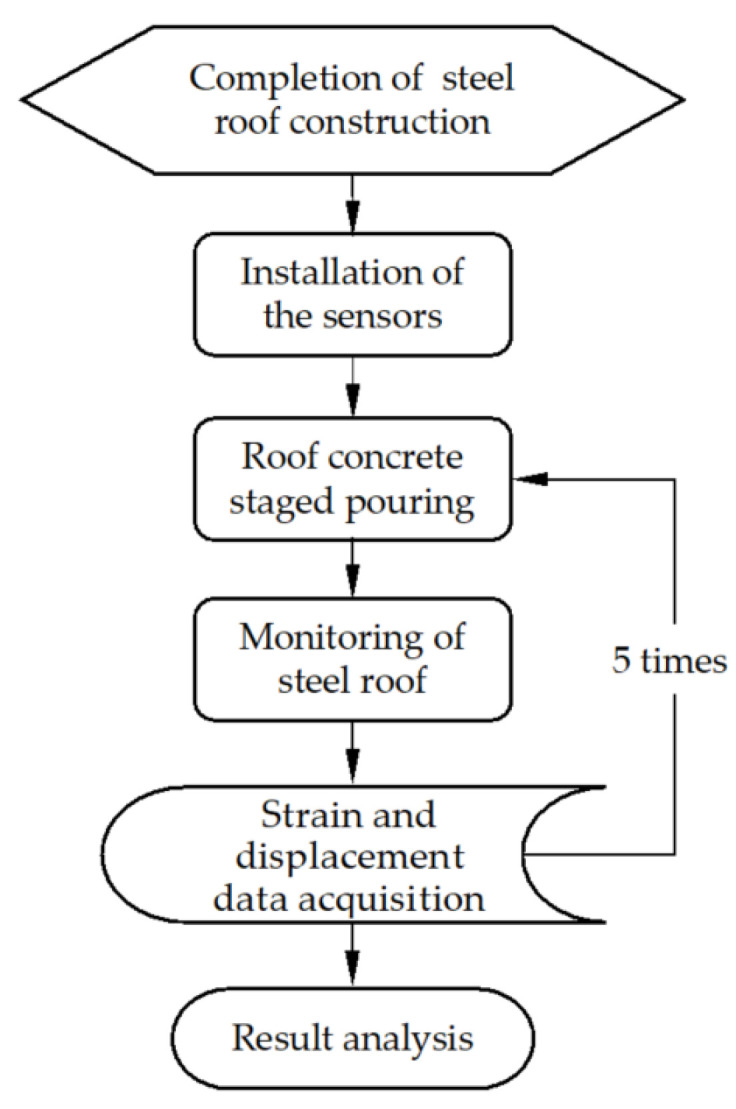
Flowchart of real-time tank monitoring.

**Figure 4 sensors-25-02498-f004:**
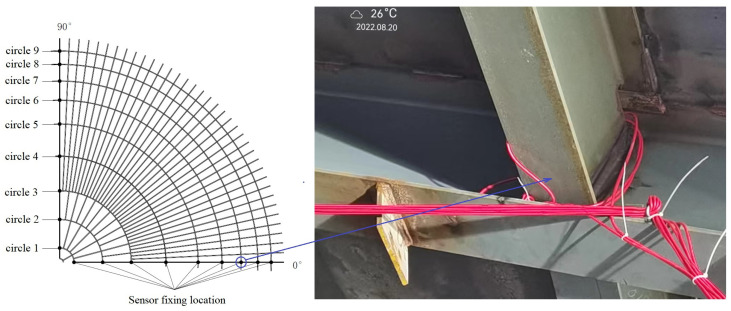
Sensor installation location.

**Figure 5 sensors-25-02498-f005:**
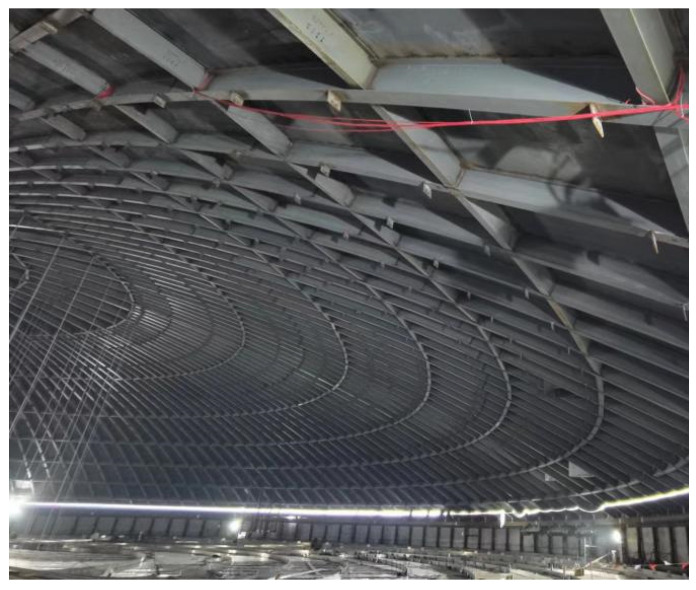
Sensor assembly line along the longitudinal beams.

**Figure 6 sensors-25-02498-f006:**
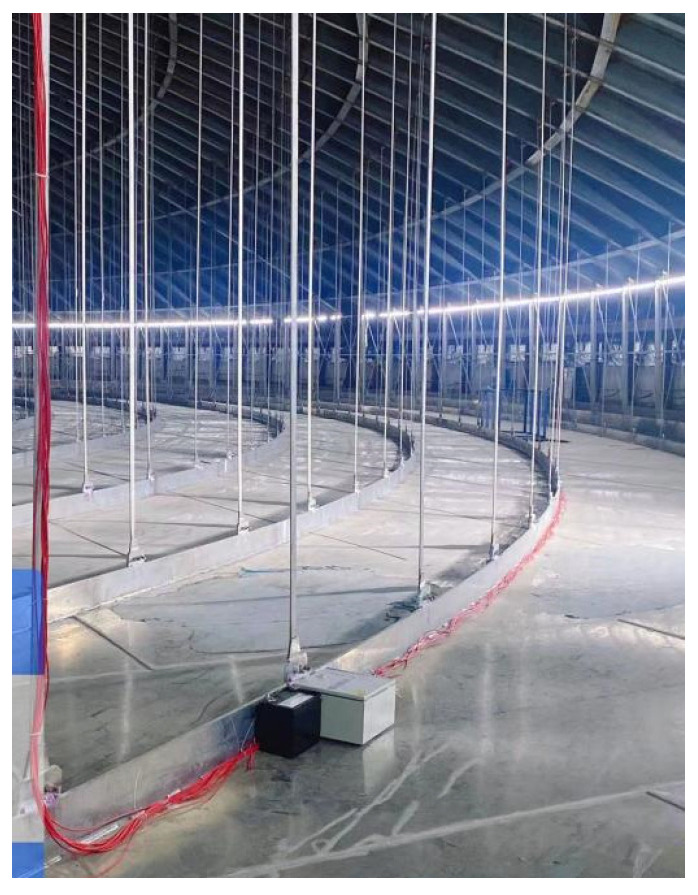
The sensor assembly line hub.

**Figure 7 sensors-25-02498-f007:**
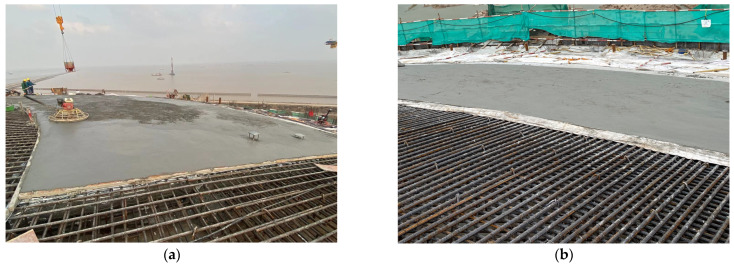
Construction photos of ring pouring. (**a**) Concrete pouring process and (**b**) completed pouring condition.

**Figure 8 sensors-25-02498-f008:**
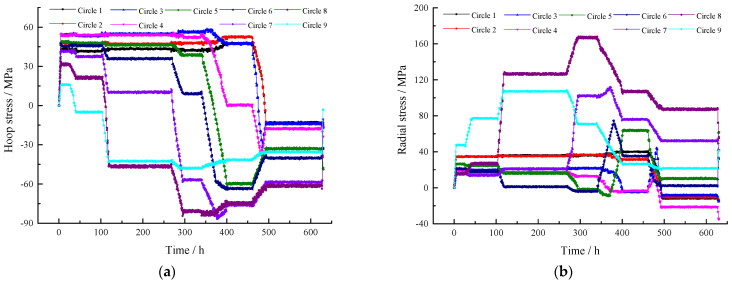
Real-time monitoring curves of stress during roof concrete construction. (**a**) Hoop stress and (**b**) radial stress.

**Figure 9 sensors-25-02498-f009:**
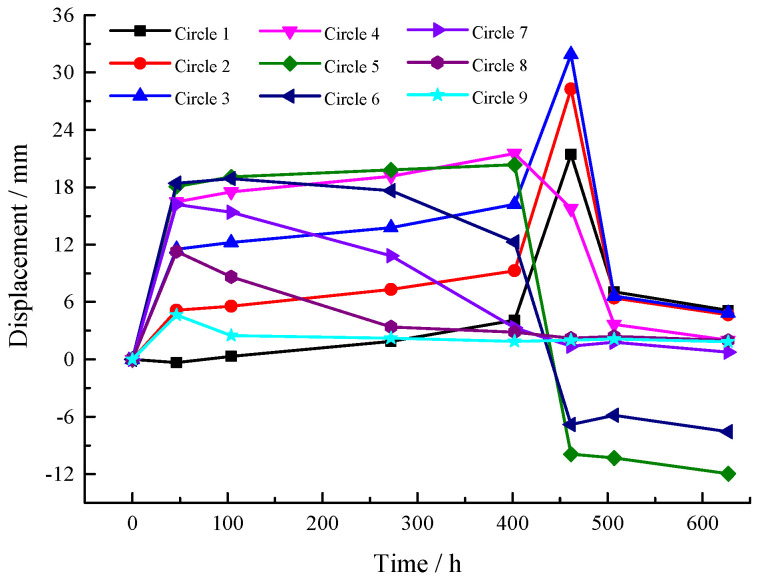
Variation in vertical displacement curves during roof concrete construction.

**Figure 10 sensors-25-02498-f010:**
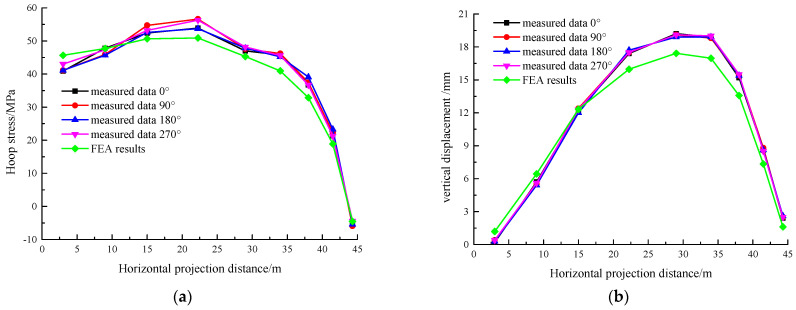
Stress and displacement results during the 1st ring pouring. (**a**) Stress and (**b**) displacement.

**Figure 11 sensors-25-02498-f011:**
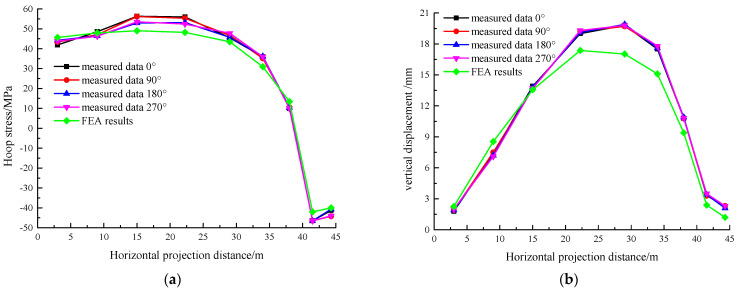
Stress and displacement results during the 2nd ring pouring. (**a**) Stress and (**b**) displacement.

**Figure 12 sensors-25-02498-f012:**
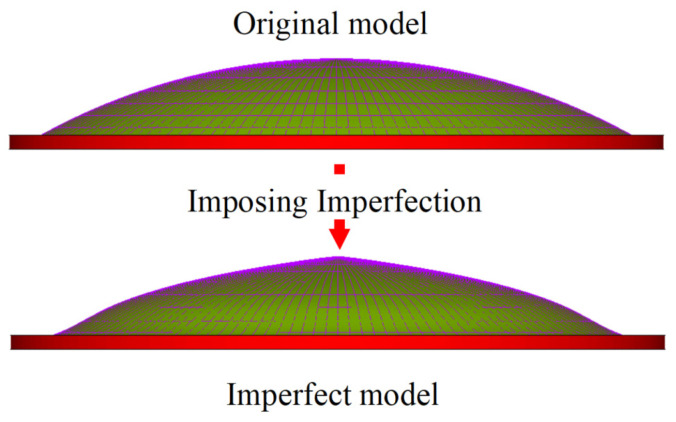
Imposing initial imperfection on model.

**Figure 13 sensors-25-02498-f013:**
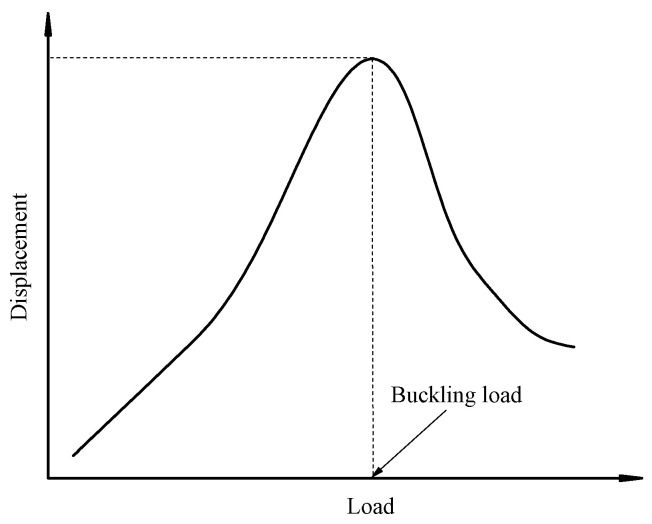
Nonlinear buckling load–displacement curve.

**Figure 14 sensors-25-02498-f014:**
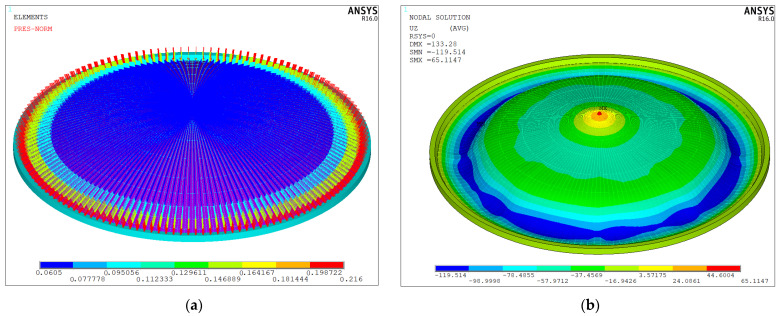
Pressure distribution and displacement contours of non-stage pouring scheme. (**a**) Pressure (unit: MPa) and (**b**) displacement (unit: mm).

**Figure 15 sensors-25-02498-f015:**
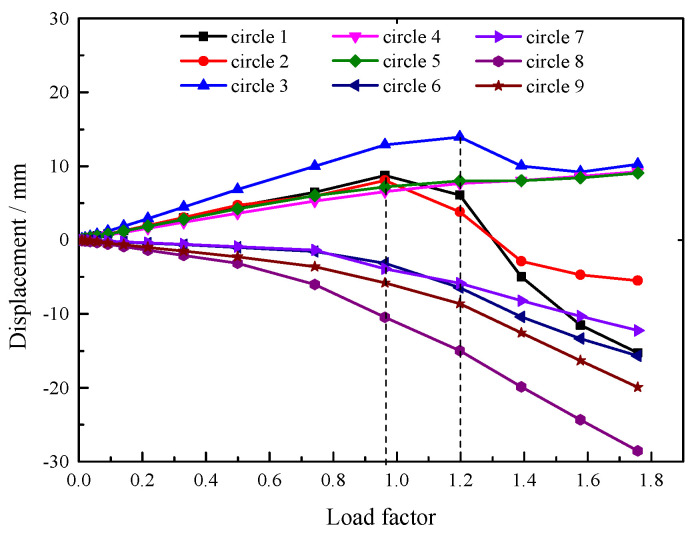
Nonlinear buckling load–displacement curve of non-stage pouring scheme.

**Figure 16 sensors-25-02498-f016:**
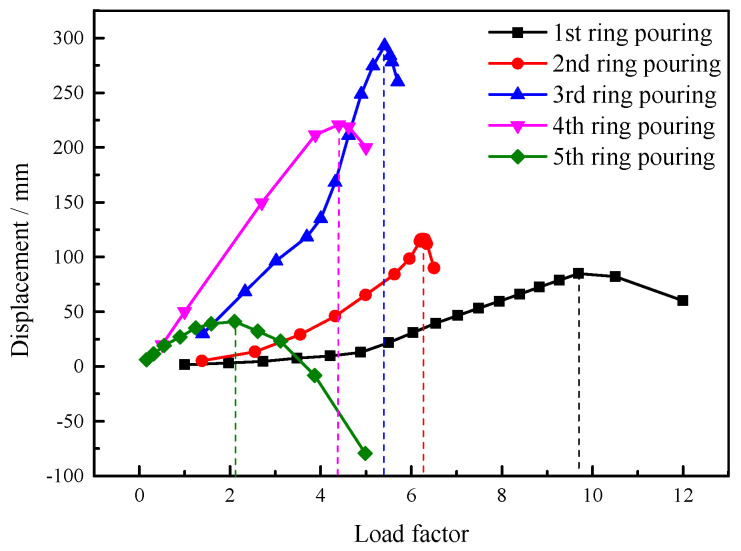
Nonlinear buckling load–displacement curve of staged pouring scheme.

**Figure 17 sensors-25-02498-f017:**
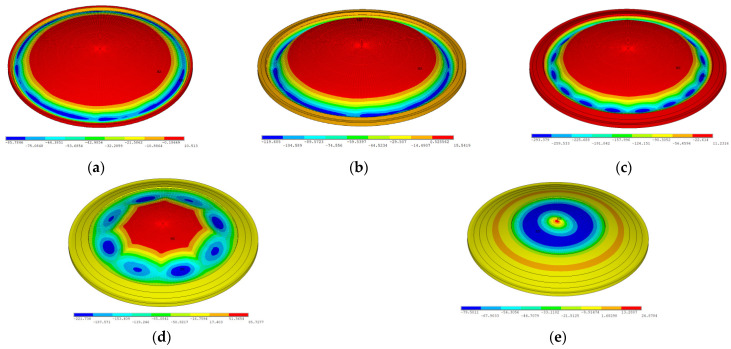
Vertical displacement contours of every ring pouring (unit: mm): (**a**) 1st ring, (**b**) 2nd ring, (**c**) 3rd ring, (**d**) 4th ring, and (**e**) 5th ring.

**Figure 18 sensors-25-02498-f018:**
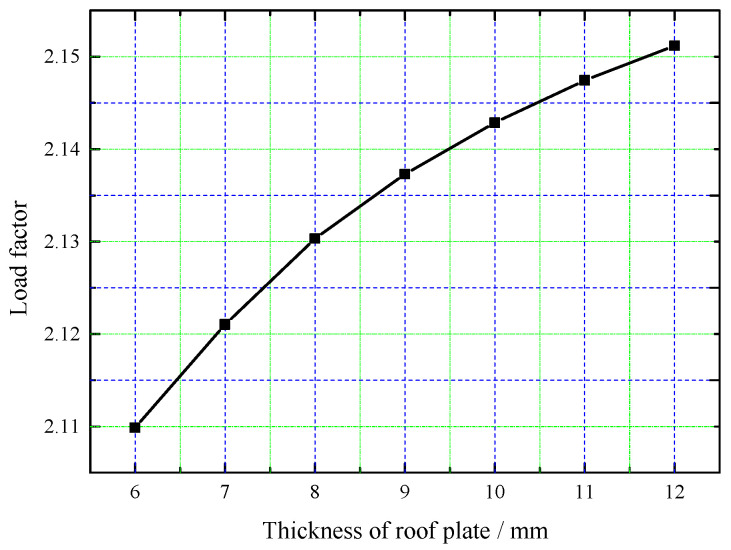
Load factor variation curve with plate thickness.

**Figure 19 sensors-25-02498-f019:**
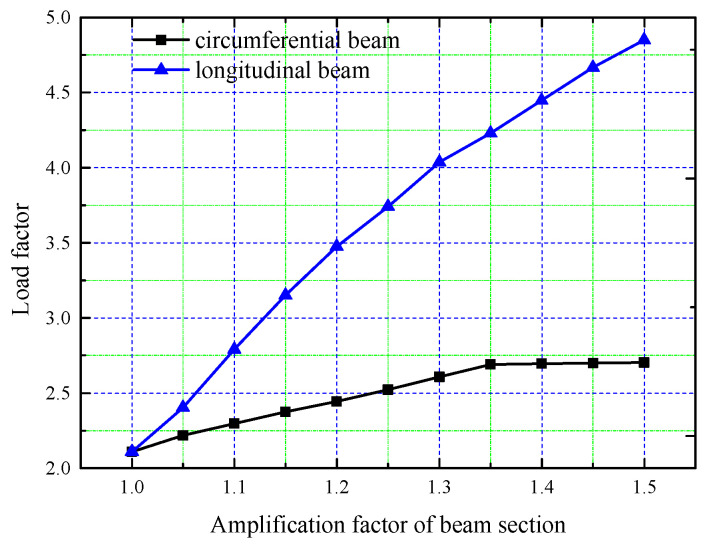
Load factor variation curve with beam dimensions.

**Figure 20 sensors-25-02498-f020:**
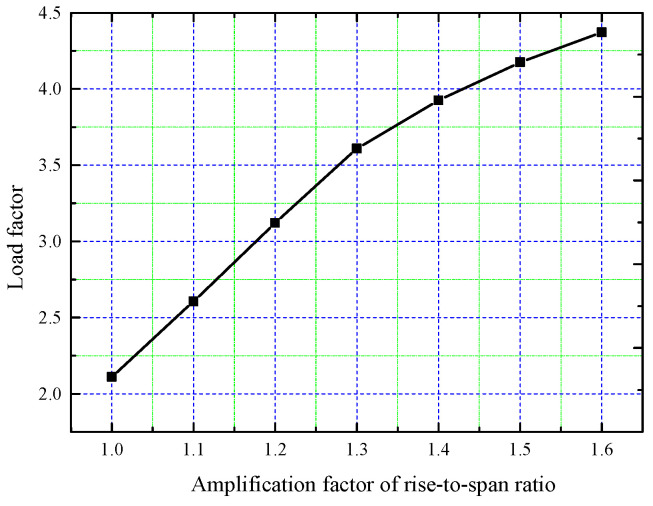
Load factor variation curve with rise–span ratio.

**Table 1 sensors-25-02498-t001:** The material properties of the roof components.

Component	Material Grade	Elastic Modulus/MPa	Poisson’s Ratio	Compressive Yield Stress/MPa	Tensile Yield Strength/MPa
Roof beams	S355 J2	206,000	0.3	355	355
Roof plate	S275 J2	206,000	0.3	275	275
Roof concrete	C50	34,500	0.2	32.4	2.64

**Table 2 sensors-25-02498-t002:** Roof concrete pouring scheme.

No. of Rings	Pouring to Radius/m	Pouring Width/m	Concrete Self-Weight Pressure/kPa
0	46	/	/
1	44.3	1.7	43.2
2	41.5	2.8	30.2
3	38	3.5	20.0
4	25	13	12.4
5	0	25	12.1

**Table 3 sensors-25-02498-t003:** Increase in buckling load factor and material consumption for different beam section size amplification factors.

Amplification Factor	1	1.1	1.2	1.3	1.4	1.5
Increase ratio of buckling load/%	0	32.34	64.75	91.33	110.75	129.88
Increase ratio of material consumption/%	0	7.95	16.65	26.11	36.32	47.30

**Table 4 sensors-25-02498-t004:** Increase in buckling load factor and material consumption for different rise–span ratio amplification factors.

Amplification Factor	1	1.1	1.2	1.3	1.4	1.5	1.6
Increase ratio of buckling load/%	0	23.48	47.88	71.03	86.02	97.86	107.2
Increase ratio of material consumption/%	0	0.87	1.83	2.85	3.94	5.10	6.33

## Data Availability

The research data of this article can accessed via the following: Mendeley Data, V1, https://doi.org/10.17632/fchkfrmxv3.1.

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
