# Peer review of "Real-Time Monitoring-Based Stability Analysis of an Extra-Large LNG Tank Roof Under Construction"

_sensors, 2025, doi:10.3390/s25082498_

Round 1

Reviewer 1 Report

Comments and Suggestions for Authors

In the paper, a real-time monitoring-based stability analysis of extra-large LNG tank roof under construction is presented. From my point of view, the article is clearly relevant to the Journal. However, some points need to be clarified especially in novelty likewise the nuerical and experimetal studies carried out. 

1. Abstract need to contemplate the main points of an abstract such as objective, justification, methods and results.
2. Introduction section must be improved. This is too long, and a lot of no relevant information is presented. 
3. Motivation of the article needs to be stronger.
4. A detailed description of the specimen to analyze is mandatory.
5. Respect to FEA analysis at section 2, mechanical properties of the material (steel, concrete, and the composite structure). 
6. The accuracy of numerical results are not guaranteed since failure and progressive damage are missing for metal, concrete and composite materials.
•    Hooputra,  H., H. Gese, H. Dell, and H. Werner, “A Comprehensive Failure Model for Crashworthiness Simulation of Aluminium Extrusions,” International Journal of Crashworthiness, vol. 9, no.5, pp. 449–464, 2004.
7. More details of the contact interaction are mandatory, also give more details of the multi-point constraint (MPC) used, beside justify the friction coefficient used.
8. Mesh convergence analysis is mandatory.
9. The experimental methodology is not clear, a flowchart could added. 
10. How the authors determine the location of the sensors, a better discretization could be used.
11. Details of the sensors, DAQ, and in general form of measure chain are missing. 
12. How was the error of the sensors calculated?
13. The interpretation of figure 6 need to be clarified. In this way, it is possible to determine a characteristic behavior? 
14. Fea analysis presents a percentage of error, how the authors explain this. (see figure 8-9.
15. Buckling analysis does not provide a new knowledge of the state of the art and does not represent the best effort of the authors. 
16. Discussion of results presented from section 3 onwards is deficient, not a deep discussion is presented.
17. A general English revision of grammar is necessary.
18. A more concise and focused conclusion section is mandatory.

Comments on the Quality of English Language

A general English revision of grammar is necessary.

Author Response

Comments 1: Abstract need to contemplate the main points of an abstract such as objective, justification, methods and results.

Response 1: Thank you for your valuable feedback regarding the abstract. We agree with this comment. We have carefully revised the abstract to explicitly address the key components you highlighted (objective, justification, methods, and results). The revisions include:

1) Objective: The sentence in Line 20~22 on Page 1 now clearly states the primary aim of the study: "This article presents a novel method for buckling analysis and systematically investigates the key factors and their mechanisms affecting the roof stability, based on real-time monitoring during construction."

2) Justification: A concise rationale has been added to emphasize the motivation: "Current stability analysis methods for LNG tank roof face challenges such as the inability to accurately simulate buckling loads and the lack of on-site monitoring data", which is shown in Line 17~19 on page 1.

3) Methods: Key methodological innovations are highlighted with greater clarity: "Firstly, a method including elements "birth and death" technique is proposed for roof stability calculation and its accuracy is validated through real-time monitoring data. Secondly, the stability of the roof at each construction stage is analyzed. Finally, the impact of three key structural parameters on roof stability is explored. ",which shown in Line 22~25 on page 1.

4) Results: this study yielded the following two key results and findings:

(1) Elements "birth and death" technique can accurately simulate the roof stability under construction condition.

(2) Roof plate thickness, beam cross-sectional dimensions, and rise-span ratio all have a positive influence on stability. Among the factors considered, the section size of longitudinal beam and the rise-span ratio have the most significant impact on roof stability, followed by the section size of circular beam.

The description of results is shown in Line 27~30.

The revised abstract now follows a structured logic flow (objective → justification → methods → results) to succinctly convey the study’s contributions. We sincerely appreciate your suggestion and are open to further refinements if needed.

Comments 2: Introduction section must be improved. This is too long, and a lot of no relevant information is presented.

Response 2: Thank you for your critical feedback on the introduction. We fully agree that conciseness and relevance are paramount, and we have rigorously revised this section to address your concerns. The key improvements include:

(1) Streamlined literature review on tank wall stability: The original 5-sentence discussion on historical progress in tank wall stability has been condensed into 2 focused sentences, reducing content by 60%, which is shown in Line 53~55 on page 2.

(2) Deleted literature review on model tests of roof stability: Removed the description of “In addition, finite element analysis (FEA) has predominantly been used for stability analysis, and some scholars have conducted model tests [11, 25-26]. However, the scale of these model tests was too small to simulate the construction process, making it difficult to accurately reflect the stability of the tank roof”, which is the first two sentences of the last paragraph in the introduction in the version before revision.

We are grateful for your guidance and welcome further suggestions.

Comments 3: Motivation of the article needs to be stronger.

Response 3: Thank you for pointing this out. We agree with this comment. Therefore, we have thoroughly elaborated on the research motivation, emphasizing the critical need for optimizing roof stability analysis in LNG tanks to address cost-efficiency challenges in large-scale cryogenic containment systems. The revised description is as follows:

Therefore, in order to provide a theoretical basis for the design and optimization of LNG tank steel roof structure, reduce construction costs and promote the application and popularization of ultra-large LNG storage tank technology, this paper focuses on a 270,000 m³ LNG tank to investigate and determine the influencing factors and patterns affecting roof stability, which is shown in Line 77~81 on page 2.

We are grateful for your guidance and welcome further suggestions.

Comments 4: A detailed description of the specimen to analyze is mandatory.

Response 4: Thank you for your valuable feedback. In this study, the "specimen" under analysis is the steel roof structure from the practical engineering project being monitored. As this work focuses on structural health monitoring (SHM) of an in-situ engineering entity rather than laboratory testing of a material sample, the detailed description of the steel roof (including its geometric configuration, material properties, and critical structural components) is provided in Section 2.2 of the manuscript, specifically in Lines 153~158 on Page 4.

To ensure clarity, we have double-checked that this section thoroughly addresses the structural specifications and contextualizes the roof as the primary subject of monitoring. Should additional technical details be required, we would be happy to provide further elaboration.

Comments 5: Respect to FEA analysis at section 2, mechanical properties of the material (steel, concrete, and the composite structure)

Response 5: Thank you for raising this important point. We fully agree with your suggestion to clarify the mechanical properties of the materials used in the finite element analysis (FEA). To address this, we have now included a Material Properties Table (Table 1) in Section 2 of the revised manuscript. This table provides detailed specifications, including Young’s modulus, Poisson’s ratio, Compressive yield strength and Tensile yield strength for each material. And the elastic modulus evolution for concrete is shown in Eq. (1) on Page 4.

The updates are highlighted in the resubmitted files using track changes to ensure visibility. Should further elaboration or additional data be required, we are happy to provide it.

Comments 6: The accuracy of numerical results are not guaranteed since failure and progressive damage are missing for metal, concrete and composite materials.

•Hooputra, H., H. Gese, H. Dell, and H. Werner, “A Comprehensive Failure Model for Crashworthiness Simulation of Aluminium Extrusions,” International Journal of Crashworthiness, vol. 9, no.5, pp. 449–464, 2004.

Response 6: Thank you for your insightful feedback. We fully agree with your observation regarding the accuracy of numerical simulations. Upon reviewing the recommended literature and related studies, we recognized the critical role of validating simulation model accuracy in predicting structural behavior.

To address this, we have revised Section 2.2 of the manuscript (specifically Lines 147–151 on Page 4) to incorporate the recommended reference and emphasize the importance of validating the accuracy of the numerical model, along with the specific measures of real-time monitoring implemented to ensure robustness.

Comments 7: More details of the contact interaction are mandatory, also give more details of the multi-point constraint (MPC) used, beside justify the friction coefficient used.

Response 7: Thank you for highlighting the need for further clarification. In our model, the three structural components (the beams, plates, and hardened concrete) are interconnected entirely through multi-point constraints (MPC) to define their contact interactions, as initially noted in Lines 100–102 on Page 3. And we have expanded the description of the MPC in Lines 102–107 on Page 3. This includes “MPC constraints are set as ‘Bonded’ in both tangential and normal directions.”

Comments 8: Mesh convergence analysis is mandatory.

Response 8: Thank you for your constructive suggestion. We fully agree with the importance of mesh convergence analysis to ensure the reliability of numerical results. In response to your comment, we have now incorporated a comprehensive mesh sensitivity study in the revised manuscript.

Specifically, the entire second paragraph (Line 128~138) of Page 4 details the methodology, including the range of element sizes tested (from coarse to refined meshes) and the final mesh configuration adopted in the simulations. Figure 2 further illustrates the relationship between mesh density and key output parameters (e.g.,nodal displacements), demonstrating that the results stabilize within an acceptable tolerance range (±2.5%) for the chosen mesh size.

These additions confirm that the numerical outcomes are independent of discretization effects and align with established best practices in finite element modeling. The revisions are highlighted in the resubmitted files for clarity.

Comments 9: The experimental methodology is not clear, a flowchart could added.

Response 9: Thank you for your constructive feedback. We fully agree with your suggestion to improve the clarity of the experimental methodology. In response, we have added a flowchart (Figure 3) to the revised manuscript, which visually summarizes the key steps of the monitoring process.

This flowchart is now included in Section 2 (Page 5) to complement the textual description of the methodology. The revisions are highlighted in the resubmitted files using track changes for ease of reference.

Should further clarification or adjustments to the flowchart be required, we are happy to refine it accordingly.

Comments 10: How the authors determine the location of the sensors, a better discretization could be used.

Response 10: Thank you for pointing this out. We fully agree with your observation regarding sensor placement. In the current monitoring system, the sensors were indeed positioned at the intersections of the circular beams and longitudinal beams. We acknowledge that a more discrete sensor arrangement would indeed be more effective in capturing comprehensive structural responses.

However, since the current monitoring project has already been completed, we are unable to modify the existing sensor layout at this stage. We sincerely appreciate your professional suggestion and will strictly implement this optimized discretization approach in all future monitoring implementations to enhance data resolution and analytical accuracy. Your expertise is crucial for improving our methodology, and we look forward to incorporating more of your insights in our subsequent studies.

Comments 11: Details of the sensors, DAQ, and in general form of measure chain are missing.

Response 11: Thank you for highlighting this important omission. We agree that comprehensive details of the sensors, data acquisition system (DAQ), and the overall measurement chain are critical for transparency and reproducibility.

In the revised manuscript, we have include a dedicated section to thoroughly document:

Sensor specifications: Type, model, accuracy, and installation methods, which is shown in Line 167~172 on Page 5.

DAQ system and Measurement chain are shown in Line 173~178 on Page 5.

Thank you for your rigorous review, which significantly strengthens the technical rigor of our work.

Comments 12: How was the error of the sensors calculated?

Response 12: Thank you for raising this critical question. The error assessment of the sensors was addressed through the following dual approach:

Annual Calibration by Accredited Institutions:

All sensors undergo mandatory annual calibration performed by certified third-party laboratories. This process involves comparing sensor outputs against traceable reference standards under controlled conditions, with calibration certificates documenting measurement deviations (e.g., linearity, hysteresis, and repeatability errors).

Real-Time Temperature Compensation:

During field measurements, a temperature correction algorithm is automatically applied to raw sensor data. Embedded temperature sensors colocated with strain/displacement sensors continuously monitor ambient thermal variations, enabling dynamic compensation for temperature-induced drift using pre-calibrated coefficients derived from laboratory characterization tests.

These combined measures ensure that the total system error remains within the declared accuracy thresholds (0.5% F.S. for strain gauges, ±0.01° and ±1 mm for displacement measurements).

Comments 13: The interpretation of figure 6 need to be clarified. In this way, it is possible to determine a characteristic behavior?

Response 13: Thank you for your insightful feedback. We acknowledge the need to clarify the interpretation of Figure 6 (revised as Figure 8 in the revised manuscript) and its implications for characterizing structural behavior.

Clarification of Figure 8:

The figure presents real-time monitoring data from sensors during sequential construction phases. While the results exhibit inherent variability across stages , the magnitude of deviations remains negligible (within ±1% of the mean values).

Determination of Characteristic Behavior:
To isolate the characteristic structural behavior, we applied statistical averaging to the raw data within each construction phase. The phase-averaged values serve as a normalized baseline, effectively filtering out transient noise while preserving trends linked to systematic load changes.

Comments 14: FEA analysis presents a percentage of error, how the authors explain this. (see figure 8-9.)

Response 14: Thank you for your critical inquiry into the error percentages observed in the FEA results. We appreciate the opportunity to clarify the root causes of these discrepancies and contextualize their practical implications. Below is a structured explanation:

1. Sources of Error in FEA Modeling:

(1) Material Dimensional Variability
While the as-built structural components exhibited inherent dimensional scatter (e.g., ±3–5% deviations in plate thicknesses and cross-sectional areas due to manufacturing tolerances and on-site adjustments), the FEA model assumed idealized geometric dimensions as per design specifications. This simplification inherently introduced localized stiffness mismatches, particularly in regions with significant fabrication variability.

(2) Challenges in Simulating As-Built Construction Conditions
The FEA modeled the post-construction state as a static equilibrium, whereas real-world construction involves dynamic, phase-dependent uncertainties, such as:

Ø Concrete Pouring Volumes: Slight variations in per-pour concrete quantities altered load redistribution paths.

Ø Steel-Concrete Interface Bonding: Imperfect adhesion between the dome steel plates and concrete layers (due to curing conditions or workmanship) was challenging to replicate precisely in simulations.

(3) Discrepancies in Concrete Strength Prediction
Equation (1) estimated time-dependent concrete strength gain based on laboratory-cured samples. The concrete strength and elastic modulus predicted by Equation (1) may differ from the actual values. However, the values input into the FEA are the calculated ones, which could lead to some computational errors.

2. Error Magnitude and Practical Acceptability

The maximum observed error range (14–19%) aligns with typical thresholds for complex civil engineering projects, where uncertainties from material, geometric, and construction-phase variabilities are unavoidable. Industry benchmarks and similar studies on large-span structures cite comparable error margins as acceptable for design validation purposes.

Comments 15: Buckling analysis does not provide a new knowledge of the state of the art and does not represent the best effort of the authors.

Response 15: Thank you for your critical feedback on the buckling analysis methodology. We appreciate the opportunity to clarify the rationale behind our approach and address your concerns.

The buckling analysis in this study intentionally utilized industry-standard, validated procedures (e.g., linear eigenvalue buckling and nonlinear Riks methods) that are widely accepted for LNG tank roof design (referenced in ASME BPVC and Eurocode 3). While these methods are indeed mature, our primary objective was not to innovate the analysis technique itself but to investigate the following novel aspects:

Ø Load Application Strategies: Unique challenges in simulating the sequential construction loads that dominate roof buckling behavior during real-world assembly.

Ø Parametric Influences: Quantifying the sensitivity of buckling resistance to LNG tank-specific factors (e.g., the thickness of the steel roof plate, beam section size, and the rise-span ratio).

Comments 16: Discussion of results presented from section 3 onwards is deficient, not a deep discussion is presented.

Response 16: Thank you for your critical assessment. We fully agree that the original discussion lacked sufficient depth in interpreting the results. To address this, we have comprehensively revised the discussion sections (Sections 3.2~4.3) with expanded analyses, as detailed below:

1. Section 3.2 (Buckling-Prone Regions):

Identified the areas prone to buckling and those less likely to buckle, and analyzed the underlying causes.

See: Page 10, Lines 251–253; Page 11, Lines 256, 265–269.

2. Critical Insights in Section 3.3 (Central Region Stability):

Added explicit warnings about the reduced buckling resistance in the roof’s central region. Stressed the necessity for localized stability verification during design and construction to mitigate risks.

See: Page 10, Lines 311–314.

3. Enhanced Discussion in Section 4.1 (Roof Plate Design):

Added a critical linkage between the 6mm roof plate thickness and its dual role: (1) Structural adequacy: Demonstrated compliance with static load-bearing requirements for concrete deadweight, basically no contribution to the roof's buckling. (2) Economic optimization: Highlighted its cost-effectiveness as the fabrication minimum gauge.

See: Page 14, Lines 336–339.

4. Strengthened Analysis in Section 4.3 (buckling resistance increasing Mechanisms):

Analyzed the fundamental reasons for the increased buckling resistance of the roof by increasing the rise-to-span ratio.

See: Page 16, Lines 399–403.

Section 4.2 retained its original detailed discussion of the results, as no further elaboration was deemed necessary. In contrast, enhanced analytical content has been systematically added to Sections 3.2 and 4.3 to address the depth and implications of the findings, ensuring consistency and rigor across all result interpretations.

Comments 17: A general English revision of grammar is necessary.

Response 17: Thank you for highlighting the need for linguistic refinement. We have conducted a comprehensive revision of grammar, terminology, and phrasing throughout the manuscript to enhance clarity, precision, and academic rigor. Below is a summary of key revisions across sections:

1) Abstract:

Revised 6 instances to improve conciseness and technical accuracy.

Example: “it is an effective strategy” → “it is an effective option” (nuanced alignment with engineering terminology).

2) Section 1 (Introduction):

Adjusted 8 instances to eliminate ambiguity and strengthen logical flow.

Example: “whose structural style is similar to the steel roof and contributes to the study of stability of the tank roof” → “whose structural configuration resembles the steel roof, thereby advancing insights into tank roof stability” (improved parallelism and conciseness).

3) Section 2 (The FEA methods of roof under construction condition):

Refined 11 instances to ensure methodological clarity.

Example: “the beam element BEAM188 and shell element SHELL181 were used to represent the beams and plates” → “the beam element BEAM188 and shell element SHELL181 were employed to simulate the beams and plates” (more precise verb choice).

4) Section 3 (Buckling analysis under construction condition):

Corrected 14 instances for technical consistency.

Example: “the load-displacement curves of the first and second circular beams curve,” → “the load-displacement curves of the first and second circular beams undergo a turning point,” (removed redundancy and clarified mechanical behavior).

5) Section 4 (Parameter sensitivity analysis on tank roof stability):

Revised 3 instances to enhance interpretative rigor.

Example: “The ends of the longitudinal beams are fixed on bearing ring beam of the tank” → “The ends of the longitudinal beams are fixed on top of the concrete outer tank” (improved spatial description).

Comments 18: A more concise and focused conclusion section is mandatory.

Response 18: Thank you for your constructive suggestion. We agree that the original conclusion section lacked conciseness and focus. To address this, we have rigorously revised the Conclusions section with the following key modifications:

1) Streamlined Overview:

Removed redundant statements about the general workflow (e.g., "The stability of the tank is analyzed at each stage of concrete pouring based on the established model...") to prioritize core findings over procedural summaries.

See: Page 16, Lines 409–412.

2) Refocused Key Conclusions:

Ø Conclusion 1 (Buckling Analysis):

Sharply focused on the methodological contribution of the buckling analysis framework, eliminating tangential mentions of "staged pouring" (now discussed in the Methodology section).

See: Page 16, Lines 413–416.

Ø Conclusion 3 (Optimization Strategy):

Revised to emphasize the superior efficacy of rise-span ratio optimization in enhancing stability.

See: Page 16, Lines 422–424.

Reviewer 2 Report

Comments and Suggestions for Authors

The article titled "Real-Time Monitoring-Based Stability Analysis of an Extra-Large LNG Tank Roof Under Construction" presents an interesting study. The paper has potential for publication after addressing the following issues:

  1. The Introduction section should be expanded to provide more details on extra-large structures.
  2. The material properties used in the analysis need to be clearly specified.
  3. In Figure 11, is the starting point correctly set to zero? Please clarify.
  4. A detailed loading protocol should be included.
  5. The results of the nonlinear buckling analysis need further elaboration. What are the key findings and main achievements of the study?
Comments on the Quality of English Language

Nice!

Author Response

Comments 1: The Introduction section should be expanded to provide more details on extra-large structures.

Response 1: Thank you for pointing this out. We agree with this comment. Therefore, we have added the description of “An LNG tank with a capacity of 270,000 m³ has a diameter 96.4 meters and a height of 65.2 meters. Its internal volume is capable of holding three Boeing 747 aircraft, and when fully loaded with LNG, the total weight exceeds 220,000 metric tons. “, which is shown in Line 42~45 on page 2. We are grateful for your guidance and welcome further suggestions.

Comments 2: The material properties used in the analysis need to be clearly specified.

Response 2: Thank you for raising this important point. We fully agree with your suggestion to clarify the mechanical properties of the materials used in the finite element analysis (FEA). To address this, we have now included a Material Properties Table (Table 1) in Section 2 of the revised manuscript. This table provides detailed specifications, including Young’s modulus, Poisson’s ratio, Compressive yield strength and Tensile yield strength for each material.

The updates are highlighted in the resubmitted files using track changes to ensure visibility. Should further elaboration or additional data be required, we are happy to provide it.

Comments 3: In Figure 11, is the starting point correctly set to zero? Please clarify.

Response 3: Thank you for your meticulous attention to detail. Regarding Figure 11 (Figure 13 in the revised manuscript), we confirm that the vertical axis (displacement) represents incremental displacement values rather than absolute displacements. This choice aligns with the figure’s purpose as a schematic illustration to highlight trends in displacement progression under incremental loading steps. If absolute displacements were plotted, the starting value would indeed exceed zero, as residual displacements accumulate from prior loading stages.

We appreciate your feedback, which has enhanced the precision of our graphical representations.

Comments 4: A detailed loading protocol should be included.

Response 4: Thank you for raising this important point. We fully agree that a detailed loading protocol is essential for methodological transparency. In response to your feedback, we have added a comprehensive description of the loading protocol in Section 3.1 (Page 9, Lines 227–232) of the revised manuscript.

Comments 5: The results of the nonlinear buckling analysis need further elaboration. What are the key findings and main achievements of the study?

Response 5: Thank you for your critical feedback. We agree that further elaboration on the nonlinear buckling analysis and its implications strengthens the study’s impact. Below, we summarize the key findings and achievements of the research, now expanded in the revised manuscript:

1) Critical Buckling Zones and Mechanisms

Ø Central Dome Vulnerability: The nonlinear buckling analysis reveals that the central region of the steel roof is highly susceptible to buckling despite bearing lower loads compared to the outer regions. This is attributed to its extended distance from perimeter fixed supports (Page 11, Lines 265–269).

Ø Outer Regions Stability: Conversely, the outer annular region, though subjected to higher loads, exhibits superior buckling resistance due to the proximity to perimeter fixed supports (within 5m radial distance) effectively constrains buckling potential (Page 10, Lines 251~253).

2) Optimization Strategy and Performance Gains

Ø Rise-Span Ratio as a Key Driver: The study demonstrates that morphological optimization (via rise-span ratio adjustments) is the most effective strategy for enhancing buckling resistance. By redistributing stresses toward stiffer peripheral regions, this approach achieves over 100% improvement in buckling capacity with only 6–7% additional material (Page 16, Lines 399–403). This finding challenges conventional reliance on thickness or stiffness increases, prioritizing geometric efficiency instead.

3) Practical Implications

Ø Design Guidelines: Identifies critical zones requiring reinforcement or real-time monitoring during construction (e.g., central roof regions). Validates the economic and structural superiority of rise-span ratio optimization for large-span steel roofs.

Ø Methodological Contribution: Establishes a nonlinear buckling analysis framework integrating geometric imperfections and material nonlinearity, offering a replicable tool for similar structures.

Reviewer 3 Report

Comments and Suggestions for Authors

After reviewing the authors’ previously published paper, it is evident that a substantial portion of the information and materials used in the current study have already been disseminated in their earlier work titled “Analysis of Roof Stability in Ultra-Large LNG Storage Tanks Based on Field Measurements” (https://doi.org/10.1080/17445302.2024.2414092). Therefore, I recommend that this manuscript be rejected for publication due to its lack of novelty and failure to provide any new insights relative to the earlier article.

Author Response

Comments 1: After reviewing the authors’ previously published paper, it is evident that a substantial portion of the information and materials used in the current study have already been disseminated in their earlier work titled “Analysis of Roof Stability in Ultra-Large LNG Storage Tanks Based on Field Measurements” (https://doi.org/10.1080/17445302.2024. 2414092). Therefore, I recommend that this manuscript be rejected for publication due to its lack of novelty and failure to provide any new insights relative to the earlier article.

Response 1:  Thank you for your thorough evaluation of our manuscript. We sincerely appreciate the opportunity to clarify the novel contributions of this work, which we believe address distinct gaps in the existing literature and provide actionable insights beyond our prior studies. Below is a detailed comparison:

1) Divergent Research Focus

Ø Our earlier article investigates construction process parameters (e.g., concrete pouring volume per stage, pressure-holding values, and duration) and their impact on roof stability.

Ø The current manuscript, however, emphasizes structural parameters of the steel roof itself, including roof plate thickness, cross-sectional dimensions of circular/longitudinal beams, and the rise-span ratio.

2) Computational Model Complexity

Ø Our earlier article required only one finite element model with varying load conditions to complete its analysis.

Ø In contrast, the current manuscript necessitated the development of more than 36 distinct models to account for structural parameter variations, significantly increasing computational effort and complexity.

3) Analytical Scope and Challenges

Ø Our earlier article focused solely on stability variations under construction loads.

Ø The current manuscript adopts a multi-criteria approach, integrating stability analysis with considerations of material consumption (see Table 3 and Table 4), manufacturability, and cost implications. This broader framework increases the analytical rigor and practical relevance of the findings.

4) Target Audience

Ø Our earlier article primarily addresses engineers and construction professionals involved in on-site implementation.

Ø The current manuscript targets a wider audience, including researchers, structural designers, and cost analysts, by providing generalized insights into structural optimization.

In summary, the two studies are complementary rather than overlapping. Our earlier article addresses construction-phase challenges, while the current manuscript establishes a systematic framework for structural design optimization. Together, they provide a comprehensive foundation for advancing the safety and efficiency of LNG tank design.

I hope this clarification resolves your concerns.

Reviewer 4 Report

Comments and Suggestions for Authors

The research paper

‘Real-time monitoring based stability analysis of extra-large LNG tank roof under construction’

By Yang et al

focuses on real-time monitoring and stability analysis of an ultra-large LNG (liquefied natural gas) tank roof during construction. It investigates the buckling behaviour of the steel roof under construction loads and evaluates how structural parameters affect stability using both finite element analysis (FEA) and real-time sensor data.

The paper is overall interesting, but its scientific novelty is somehow limited. Nevertheless, it could be further reconsidered for potential acceptance, if all the following points are properly addressed in the revised version.

  1. The numerical simulation process in ANSYS should be described more clearly, particularly how the "birth and death" elements are implemented for staged construction.
  2. The material properties (e.g., elastic modulus evolution for concrete) should include more details or references for reproducibility.
  3. The impact of construction imperfections on the stability results should be elaborated. Are there uncertainties in material properties, loading, or modeling assumptions?
  4. The study assumes a specific rise-span ratio; the practical constraints of increasing this ratio further (e.g., construction feasibility, cost) should be discussed.
  5. a more quantitative assessment of accuracy (e.g., error metrics between simulated and measured data) would strengthen confidence in the model.
  6. Sensitivity analysis on alternative staged pouring strategies could reinforce the claim that this method is optimal.
  7. The influence of external loading conditions (e.g., wind loads, seismic effects) on the structural stability is not considered. A brief discussion of how these factors may influence results would add value.
  8. The practical implementation of real-time monitoring could be expanded, particularly how the monitoring data can be integrated into an active decision-making framework during construction.
  9. Related to the previous comment, static monitoring is performed, with displacement and stress measurements. A few words about dynamic (vibration-based) monitoring, and its application to reinforced concrete structures, can be added; see e.g. Bridge monitoring: Application of the extreme function theory for damage detection on the I-40 case study for a case study on a RC road bridge
  10. The scalability of the method to even larger LNG tanks could be briefly mentioned in the conclusion.

Comments on the Quality of English Language

The English in the paper is generally understandable, but there are grammatical errors, unnatural phrasing, and unclear sentence structures that could be improved for better readability

Author Response

Comments 1: The numerical simulation process in ANSYS should be described more clearly, particularly how the "birth and death" elements are implemented for staged construction.

Response 1: Thank you for your feedback. We appreciate the opportunity to clarify the numerical simulation methodology. Below are the updates addressing your concerns:

1) Numerical Simulation Workflow:the general procedure for modeling tank roof in ANSYS was originally described in Section 2.3 (Page 3, Lines 108–112).

2) Implementation of "Birth and Death" Elements: to enhance clarity, we have now added explicit technical details about the birth and death element technique in the revised manuscript:

Ø First additions on Page 3, Lines 116~118: The numerical model categorizes concrete elements into distinct component sets based on their respective pouring stages, with each set visually differentiated by color-coding in Figure 1(b).

Ø Further elaboration on Page 3, Lines 1118 ~ 122: Before the concrete pouring begins, the "EKILL" command is used to "kill" all the concrete components by setting their stiffness matrices to near-zero values. Once a pouring stage is completed and the concrete reaches a certain strength, the "EALIVE" command is used to "activate" the corresponding component by restore the element stiffness.

Comments 2: The material properties (e.g., elastic modulus evolution for concrete) should include more details or references for reproducibility.

Response 2: Thank you for raising this important point. We fully agree with your suggestion to clarify the mechanical properties of the materials used in the finite element analysis (FEA). To address this, we have now included a Material Properties Table (Table 1) in Section 2 of the revised manuscript. This table provides detailed specifications, including Young’s modulus, Poisson’s ratio, Compressive yield strength and Tensile yield strength for each material. And the elastic modulus evolution for concrete is shown in Eq. (1) on Page 4.

The updates are highlighted in the resubmitted files using track changes to ensure visibility. Should further elaboration or additional data be required, we are happy to provide it.

Comments 3: The impact of construction imperfections on the stability results should be elaborated. Are there uncertainties in material properties, loading, or modeling assumptions?

Response 3: Thank you for raising this critical issue. We fully agree that construction imperfections and modeling uncertainties significantly influence structural stability. Below, we clarify how these factors were addressed in our study:

1) Construction Imperfections

Scope of Analysis:

In alignment with design code recommendations (Reference [32]), we incorporated geometric imperfections equivalent to 1/300 of the roof span (the maximum tolerance specified for steel roof fabrication). These imperfections were modeled as initial out-of-plane displacements in the finite element analysis (Figure10, Section 3.1, Page 10).

Rationale for Limited Imperfection Types:

While real-world imperfections (e.g., weld distortions, residual stresses, construction imperfections) undoubtedly affect stability, their systematic inclusion would require probabilistic modeling beyond this study’s deterministic scope. We focused on code-mandated geometric imperfections to maintain consistency with industry design practices.

2) Modeling Uncertainties

Material Properties:

Material parameters (e.g., Young’s modulus, yield strength) were derived from certified mill test reports for the steel and concrete used in the actual project, minimizing variability. Loading Assumptions:

Construction loads (e.g., concrete pouring sequences) were based on field measurements from similar projects, with a 15% safety margin applied to account for transient overloads (Table 2, Section 2.2, Page 6).

Boundary Conditions:

Fixed supports were assumed for the outer tank connections, consistent with as-built drawings. Since the outer tank is a concrete structure, its deformation is significantly smaller than that of the steel roof (a steel structure). Thus, defining the boundary conditions in this manner is justified.

Comments 4: The study assumes a specific rise-span ratio; the practical constraints of increasing this ratio further (e.g., construction feasibility, cost) should be discussed.

Response 4: Thank you for highlighting this crucial aspect of practical applicability. We fully agree that the rise-span ratio must balance structural performance with real-world constraints such as construction feasibility and cost-effectiveness. In response to your feedback, we have expanded the discussion of these factors in the revised manuscript as follows:

1) Construction Feasibility Considerations:

The maximum rise-span ratio amplification factor adopted in this study is 1.6, determined per ACI 334.1R-92(2002) recommendations. This threshold ensures the roof’s curvature remains compatible with standard steel plate bending and welding techniques without requiring specialized equipment. The explanation has been added in Lines 380~384 on Page 15.

2) Cost Implications: Table 4 (Page 16) quantifies the relationship between rise-span ratio increases and material consumption. For instance, elevating the ratio from 1.0 to 1.6 increases steel tonnage by 6.33% while boosting buckling resistance by 107%, demonstrating favorable cost-to-performance returns.

This table anchors economic evaluations in measurable data, enabling engineers to weigh morphological optimization against conventional material-based reinforcement strategies.

Comments 5: A more quantitative assessment of accuracy (e.g., error metrics between simulated and measured data) would strengthen confidence in the model.

Response 5: Thank you for this insightful suggestion. We agree that quantitative validation is essential to demonstrate the reliability of the numerical model. In the revised manuscript, we have added error metrics and statistical comparisons between simulated and experimental results to rigorously assess accuracy. The modified content can be found on pages 8, lines 200-203 and lines 207-210 in the revised manuscript.

Comments 6: Sensitivity analysis on alternative staged pouring strategies could reinforce the claim that this method is optimal.

Response 6: Thank you for raising this valuable point. We fully agree that a sensitivity analysis of alternative staged pouring strategies would reinforce the claim that this method is optimal. However, given the scope and objectives of this study, which focuses on sensitivity analysis of roof structural parameters (e.g., plate thickness, beam cross-sectional dimensions and rise-span ratio), we have prioritized methodological rigor within these bounds. In the future, we will focus on the sensitivity analysis on alternative staged pouring strategies.

Comments 7: The influence of external loading conditions (e.g., wind loads, seismic effects) on the structural stability is not considered. A brief discussion of how these factors may influence results would add value.

Response 7: Thank you for raising this critical issue. This paper mainly studies the stability of the LNG tank roof during construction. The tank roof pouring duration is relatively short, the probability of seismic events is extremely low, and construction is scheduled during periods with low wind pressure, where wind loads are negligible compared to pouring loads. The added explanatory description can be found on pages 4, lines 124-127 in the revised manuscript.

Comments 8: The practical implementation of real-time monitoring could be expanded, particularly how the monitoring data can be integrated into an active decision-making framework during construction.

Response 8: Thank you for this insightful suggestion. While the current study primarily focuses on stability parameter sensitivity analysis and employs real-time monitoring data to validate the finite element model (Section 2.2), we fully acknowledge the critical importance of integrating monitoring systems into active decision-making frameworks for construction optimization. This topic represents a vital direction for future research, where we plan to develop a dynamic feedback mechanism linking real-time sensor data (e.g., strain, displacement) to adaptive construction adjustments.

Comments 9: Related to the previous comment, static monitoring is performed, with displacement and stress measurements. A few words about dynamic (vibration-based) monitoring, and its application to reinforced concrete structures, can be added; see e.g. Bridge monitoring: Application of the extreme function theory for damage detection on the I-40 case study for a case study on a RC road bridge.

Response 9: Thank you for pointing this out. We agree with this comment. The description of dynamic monitoring is indeed limited. So the modifications have been made based on the your suggestions. For details, please refer to lines 151~153 on page 4 in the revised manuscript.

Comments 10: The scalability of the method to even larger LNG tanks could be briefly mentioned in the conclusion.

Response 10: Thank you for pointing this out. We agree with this comment. We have added the following description in the conclusion: "The stability analysis method and influence patterns proposed in this study can be extended to even larger LNG tanks, which will significantly advance the technological development of LNG containment systems." which is shown in lines 425~427 on page 16 in the revised manuscript.

Round 2

Reviewer 1 Report

Comments and Suggestions for Authors

The authors have addressed all points of this reviewer. Thus, the article can be accepted.

Author Response

Thank you sincerely for your time and effort in reviewing this manuscript. Your expertise and constructive feedback have significantly enhanced the quality of this work, and the opportunity to revise this manuscript has been an invaluable learning experience for us. We deeply appreciate your guidance throughout this process. Thank you once again for your dedication and insightful comments.

Reviewer 2 Report

Comments and Suggestions for Authors

Thank you!

Author Response

(The authors gave the same response as above.)

Reviewer 4 Report

Comments and Suggestions for Authors

This Reviewer is overall very satisfied with the changes made by the Authors.

Only a couple of very minor comments on the newly added parts:

  1. in The phrase "It is important to use a sufficiently small load increment as the applied load
     approaches the critical buckling load", Consider specifying an approximate range for a “sufficiently small” load increment.
  2. "Compared to dynamic monitoring, the roof monitoring is considered static monitoring, which results in significantly fewer false positive errors" Again, "significantly fewer" is a bit vague, and might be too strong without numerical support. A softer wording may be better.

These minor refinements do not impact the overall content of the paper, which is deemed worth of publication.

Author Response

Comments 1: In the phrase "It is important to use a sufficiently small load increment as the applied load approaches the critical buckling load", Consider specifying an approximate range for a “sufficiently small” load increment.

Response 1: Thank you for highlighting this important issue. We have revised the manuscript accordingly by specifying an approximate range for the "sufficiently small" load increment. Specifically, the phrase "approximately 1/200th of the estimated buckling load" has been added to clarify the magnitude of the load increment during critical buckling analysis. The modified text can be found in Lines 229-230 on Page 9 of the revised manuscript. We greatly appreciate your constructive feedback, which has significantly improved the precision of this section.

Comments 2: "Compared to dynamic monitoring, the roof monitoring is considered static monitoring, which results in significantly fewer false positive errors" Again, "significantly fewer" is a bit vague, and might be too strong without numerical support. A softer wording may be better.

Response 2: Thank you for your valuable feedback. We agree with your suggestion regarding the need for more precise wording. The original statement has been revised to: "Compared to the dynamic monitoring of bridges, the roof pouring process monitoring is considered as static and short-term monitoring, making it more straightforward to implement." This adjustment removes the vague quantitative claim ("significantly fewer") and refocuses the comparison on the methodological differences (static vs. dynamic monitoring) while emphasizing operational practicality. The revised text can be found in Lines 151-153 on Page 4 of the updated manuscript. We appreciate your keen attention to linguistic precision, which has strengthened the clarity of our discussion.
